# Fission Impossible (?)—New Insights into Disorders of Peroxisome Dynamics

**DOI:** 10.3390/cells11121922

**Published:** 2022-06-14

**Authors:** Ruth E. Carmichael, Markus Islinger, Michael Schrader

**Affiliations:** 1College of Life and Environmental Sciences, Biosciences, University of Exeter, Exeter EX4 4QD, UK; r.carmichael@exeter.ac.uk; 2Institute of Neuroanatomy, Mannheim Centre for Translational Neuroscience, Medical Faculty Mannheim, University of Heidelberg, 68167 Mannheim, Germany; markus.islinger@medma.uni-heidelberg.de

**Keywords:** peroxisomes, mitochondria, organelle dynamics, division defects, dynamin-related protein 1, mitochondrial fission factor, PEX11β, FIS1, ACBD5, membrane fission

## Abstract

Peroxisomes are highly dynamic and responsive organelles, which can adjust their morphology, number, intracellular position, and metabolic functions according to cellular needs. Peroxisome multiplication in mammalian cells involves the concerted action of the membrane-shaping protein PEX11β and division proteins, such as the membrane adaptors FIS1 and MFF, which recruit the fission GTPase DRP1 to the peroxisomal membrane. The latter proteins are also involved in mitochondrial division. Patients with loss of DRP1, MFF or PEX11β function have been identified, showing abnormalities in peroxisomal (and, for the shared proteins, mitochondrial) dynamics as well as developmental and neurological defects, whereas the metabolic functions of the organelles are often unaffected. Here, we provide a timely update on peroxisomal membrane dynamics with a particular focus on peroxisome formation by membrane growth and division. We address the function of PEX11β in these processes, as well as the role of peroxisome–ER contacts in lipid transfer for peroxisomal membrane expansion. Furthermore, we summarize the clinical phenotypes and pathophysiology of patients with defects in the key division proteins DRP1, MFF, and PEX11β as well as in the peroxisome–ER tether ACBD5. Potential therapeutic strategies for these rare disorders with limited treatment options are discussed.

## 1. Introduction

Organelle dynamics refers to the ability of subcellular, membrane-bound organelles to alter their morphology, size, abundance, motility and interplay with other cellular compartments. Dynamic changes in morphology enable the organelles to respond to alterations in the cellular environment and to adapt to changing cellular needs. The dynamics of mitochondria are arguably best studied and are mediated by fusion and fission events, which can result in the formation of complex interconnected networks or smaller entities. Mitochondrial dynamics are mediated by a growing number of membrane-shaping proteins, but also involve other organelles and the cytoskeleton [1,2,3,4,5]. The dynamic changes in mitochondrial morphology impact mitochondrial and cellular processes, such as apoptosis/cell death, mitophagy, mitochondrial metabolism and ATP production, mitochondrial quality control and mtDNA inheritance, cell fate decisions, cell cycle, and the transport/distribution of mitochondria, to name a few [4,6,7].

Peroxisomes are also dynamic, highly plastic organelles that are able to adjust their morphology, number, intracellular position, interactions with other organelles and metabolic functions according to the needs of the cell or organism. Particular focus has been on dynamic processes involved in the growth and division/multiplication of peroxisomes [8,9]. Remarkably, peroxisomes and mitochondria share several key components of their division/fission machinery, including the membrane adaptor proteins FIS1 (Fission 1) and MFF (mitochondrial fission factor), which recruit the large dynamin-like GTPase DRP1 (dynamin-related protein 1) to the organelle membrane (reviewed in [10,11,12]). Once associated with adaptor proteins, DRP1 forms an oligomeric ring around the organelle membrane. The hydrolysis of GTP to GDP then causes a conformational change in the ring, which executes membrane fission [13,14]. Interestingly, the sharing of key division components between mitochondria and peroxisomes appears to be evolutionarily conserved amongst mammals, fungi and plants [15]. However, in contrast to mitochondria, mature peroxisomes do not appear to fuse, at least not in a way similar to mitochondrial fusion [16,17]. Mitochondrial fusion involves the large GTPases Mitofusin 1 (MFN1) and Mitofusin 2 (MFN2) at the outer mitochondrial membrane, and the GTPase OPA1 (Optical Atrophy 1), which mediates fusion of the inner mitochondrial membrane [4,18]. In contrast to the fission machinery, these key fusion components do not localize to peroxisomes [16]. There are additional membrane-shaping and adaptor proteins, which appear to be organelle specific. The peroxisomal biogenesis factor (peroxin) PEX11β is a membrane-shaping protein involved in multiple steps of peroxisomal growth and division (see Section 3.4). MiD49 and MiD51 are mitochondria-specific adaptor proteins at the outer mitochondrial membrane, which regulate mitochondrial division [19]. Peroxisomes can also form de novo under certain conditions, which involves the ER and mitochondria [20,21]. This is mediated by the formation of ER- and mitochondria-derived vesicles, which, however, does not depend on DRP1 or PEX11β [21,22].

Peroxisomes are oxidative organelles with key functions in the β-oxidation of fatty acids, synthesis of ether lipids/plasmalogens (important components of the brain white matter) and reactive oxygen species (ROS) homeostasis [23]. They show a close interplay with mitochondria, including metabolic cooperation in fatty acid β-oxidation, ROS homeostasis, and anti-viral responses [10,11,24,25]. Membrane contacts between mitochondria and peroxisomes have been identified, which link peroxisomal β-oxidation and mitochondrial ATP production [26,27,28]. The interconnection of these metabolic activities may have contributed to the sharing of division proteins, which allows for coordination of peroxisomal and mitochondrial fission under specific metabolic conditions. Interestingly, many of the membrane proteins shared by peroxisomes and mitochondria are tail-anchored (TA) membrane proteins including the DRP1-adaptors FIS1 and MFF; GDAP1 (ganglioside-induced differentiation-associated protein 1) [29], which may contribute to fission in neurons; MAVS (mitochondrial antiviral-signaling protein) [30], which is involved in anti-viral signaling; and MIRO1, which is an adaptor for microtubule-dependent motor proteins, such as kinesin [31,32,33]. TA proteins have a single transmembrane domain (TMD) and a short amino acid tail at the very C-terminus, which determine post-translational targeting to different organelles [34]. Peroxisomal targeting of TA proteins is mediated by PEX19, the receptor/chaperone for multiple peroxisomal membrane proteins [35], and is promoted by a moderate hydrophobicity of the TMD and a high positive net charge of the tail [36]. Reducing the positive net charge results in additional targeting to mitochondria; on the other hand, increasing the tail charge of mitochondrial TA proteins results in peroxisomal localization [36,37]. These observations may explain why several TA proteins are shared between peroxisomes and mitochondria: slight alterations in amino acid composition of the tail region may have occurred during eukaryote evolution, which allowed dual targeting to both organelles [25,36]. Due to the interconnection of peroxisomal and mitochondrial functions, e.g., in the β-oxidation of fatty acids in both organelles in animals, the sharing of TA adaptor proteins may have been an evolutionary advantage in coordinating organelle dynamics and cellular metabolism.

Patients with mutations in DRP1 or MFF have now been identified, showing defects in peroxisomal and mitochondrial dynamics (see Section 3.1 and Section 3.2). The patients suffer from developmental defects, neurological abnormalities and loss of sensory functions, whereas metabolic functions of the organelles are generally not or only slightly affected. This has hampered the diagnostics of those disorders, which are often based on metabolic biomarkers, but also highlights the importance of membrane dynamics for human health and development.

Here, we provide a timely update on peroxisomal membrane dynamics with particular focus on peroxisome formation by membrane growth and division. We address the role of PEX11β in these processes, as well as the role of peroxisome–ER contacts in lipid transfer for peroxisomal membrane expansion. Furthermore, we summarize the clinical phenotypes and pathophysiology of patients with defects in the key division proteins DRP1, MFF, and PEX11β as well as in the peroxisome–ER tether ACBD5. As these are rare disorders with limited treatment options, we discuss potential therapeutic strategies.

## 2. Growth and Division of Peroxisomes

A model for peroxisome biogenesis by growth and division of pre-existing organelles was first proposed in 1985 [38]. The identification and molecular characterization of peroxisomal division proteins, microscopic observations and the analysis of patient fibroblasts have contributed to a refined model (Figure 1). In mammalian cells, peroxisome formation by membrane growth and division represents a multi-step process involving the remodeling of the peroxisomal membrane, membrane expansion/elongation (growth), membrane constriction and final scission (fission) [8,9,39]. Peroxisomal growth and division results in the formation of new peroxisomes (multiplication/proliferation), which import matrix and membrane proteins to maintain functionality [40].

### 2.1. Membrane Deformation and Elongation

The peroxisomal membrane protein PEX11β has key roles in many, if not all, of the steps of the growth and division pathway (Figure 1). PEX11β possesses two membrane-spanning domains with a very short C-terminus and a larger N-terminus, both facing the cytosol [41,42] (Figure 2). The N-terminal domain contains amphipathic helices, which allow interactions with membrane lipids [43,44]. Furthermore, the N-terminus is required for the oligomerization of PEX11β [41,45,46]. Expression of PEX11β initially results in the deformation of the peroxisomal membrane at a defined site on the “mother” peroxisome. Subsequently, a membrane protrusion forms, which further elongates, before it constricts at multiple sites [40] (Figure 1). It is suggested that both phospholipid binding via the amphipathic helices and oligomerization of PEX11β are the driving forces for membrane remodeling, deformation and elongation. In addition to a PEX11β protein scaffold, certain membrane lipids, which promote changes in membrane curvature, may be required [9]. The role of the PEX11 isoforms PEX11α and PEX11γ in peroxisome division is less clear (see Section 3.4).

### 2.2. Membrane Constriction and Assembly of Fission Sites

Although DRP1 can form ring-like structures around organelle membranes, the diameter of the organelles is too large to allow ring assembly and requires constriction prior to DRP1 assembly. As peroxisomes can still constrict after loss of DRP1 [47] (see Section 3.1), other factors besides DRP1 are required. How peroxisomes constrict prior to fission is still unknown, although PEX11β may play a role as it is found at constriction sites [48], its manipulation blocks constriction [40], and it can constrict liposomes in vitro [49] (Figure 1). Mitochondrial division is facilitated by contact with the endoplasmic reticulum (ER) to form constriction sites by wrapping extended ER tubules around mitochondria [2]. DRP1 and its adaptors assemble at the mitochondria–ER contacts. These interaction sites may be determined by replicating mtDNA, which is positioned at these sites [3]. Furthermore, the ER-bound inverted-formin 2 (INF2) and the mitochondrial-anchored formin-binding Spire1C proteins assemble actin filaments at the mitochondria–ER contact sites, which mediate constriction prior to DRP1 ring formation [1,5]. Recent studies suggest that both membrane bending (induced by constriction) and tension (e.g., by cytoskeletal forces) contribute to mitochondrial fission at constriction sites [50,51]. If cooperative processes between the ER and actin are also involved in peroxisome constriction is unclear, but cannot yet be excluded. Peroxisomes are in close contact with the ER and form membrane contacts [52,53]. However, the ER is mainly found at the mother peroxisomes, and not at peroxisome tubules, which would need to constrict [54]. A major role of the peroxisome–ER contacts is the supply of membrane lipids for peroxisomal membrane expansion [31,52,55] (see Section 3.3). Mitochondrial constriction may be more complex than peroxisome constriction and require additional forces, as mitochondria need to coordinate division of their outer and inner membranes and may link this to mtDNA replication. As peroxisomes do not contain DNA and have only a single limiting membrane, constriction and division processes may be less complex.

### 2.3. Membrane Scission

Constriction of the peroxisomal membrane goes along with the assembly of the division machinery. It is composed of PEX11β and the adaptor proteins FIS1 and MFF, which can recruit DRP1 to the peroxisomal (and mitochondrial) membrane (Figure 1). Both FIS1 and MFF have been shown to interact with PEX11β at peroxisomes [42,46,56,57]. PEX11β is not believed to be an adaptor for the recruitment of DRP1, but can interact with DRP1 to promote DRP1 assembly and subsequent stimulation of its GTPase activity [58]. DRP1 activity depends on a supply of GTP, which may be locally generated by DYNAMO1 (dynamin-based ring motive-force organizer 1)/NME3 (nucleoside diphosphate kinase 3). The *Cyanidioschyzon merolae* DYNAMO1 protein localizes to both the peroxisomes and mitochondria with a role in fueling membrane fission through local GTP generation [59]. NME3, the mammalian orthologue, was also found to localize to peroxisomes. NME3-suppression results in elongated peroxisomes, suggesting a role in peroxisome division by efficiently loading DRP1 with GTP and thus facilitating its fission activity [60].

The role of FIS1 in peroxisome division has been controversial [56,61,62]. Whereas the loss of MFF results in highly elongated peroxisomes (and mitochondria) due to a block in peroxisome division (see Section 3.2), the loss of FIS1 does not appear to change peroxisome morphology. This has led to the assumption that MFF is the major adaptor protein for peroxisomal (and mitochondrial) fission, whereas FIS1 may fulfill more specialized functions [62]. However, we have recently shown that overexpression of PEX11β can promote peroxisome division in MFF-deficient fibroblasts dependent on FIS1 [63]. Furthermore, overexpression of MFF in PEX11β-deficient fibroblasts restored the normal, spherical peroxisome morphology. These findings indicate that two independent mechanisms for peroxisome division may exist, one via MFF and another via PEX11β/FIS1 (see Section 5).

### 2.4. Pulling Forces, ER Contacts and Lipid Transfer

After division, newly formed peroxisomes need to be distributed within the cell (Figure 1). Peroxisomes move along microtubules in mammalian cells and can recruit microtubule-dependent motor proteins, such as kinesin and dynein [64,65,66]. The Rho GTPase MIRO1 has been shown to localize to peroxisomes and mitochondria [31,32,33]. Like FIS1 and MFF, MIRO1 is also a C-tail anchored membrane protein with dual peroxisomal and mitochondrial localization [33,36]. It can serve as an adaptor protein for the recruitment of microtubule motor proteins [67]. The MIRO1/motor complex can exert pulling forces at peroxisomes, which can lead either to peroxisome motility along microtubules and re-positioning, or to the formation of membrane protrusions/membrane expansion [31]. The latter process requires attachment of peroxisomes to a fixed point, e.g., to the cytoskeleton or to the ER. Although the depolymerization of microtubules curiously promotes peroxisomal elongation and does not inhibit fission [68,69], microtubule-dependent pulling forces via MIRO1/motor proteins may facilitate peroxisomal membrane expansion and division [31].

Expression of a peroxisome-targeted MIRO1 (to avoid mitochondrial alterations) promoted the formation of peroxisomal membrane protrusions in PEX5-deficient patient fibroblasts [31]. PEX5 is a cytoplasmic import receptor for peroxisomal matrix proteins, which can interact with the peroxisomal membrane to deliver cargo proteins [70,71]. Loss of PEX5 function (or of other peroxins of the peroxisomal matrix protein import machinery) causes Zellweger syndrome (ZS), a spectrum of peroxisome biogenesis disorders with severe developmental and neurological defects [72]. It results in import-deficient peroxisomes, which lack peroxisomal enzymes within the matrix and are metabolically inactive [73]. However, peroxisomal membranes can still be formed, but peroxisomes are reduced in number and enlarged, presenting as “ghosts” (empty peroxisomal membrane structures) [74]. Interestingly, these peroxisomal “ghosts” are still dynamic, as they can form membrane protrusions promoted by MIRO1/motor proteins and can also elongate and divide, e.g., after PEX11β expression [31]. This indicates that these peroxisomal membrane dynamics are mechanistically independent of peroxisomal metabolism. Patients with a defect in peroxisomal β-oxidation, e.g., in acyl-CoA oxidase 1 (ACOX1), also exhibit enlarged peroxisomes in skin fibroblasts. Remarkably, addition of docosahexaenoic acid (DHA, C22: 6n-3) restored the normal peroxisome morphology in those cells [45]. The synthesis of DHA requires the cooperation of peroxisomes and the ER; the precursor undergoes one round of β-oxidation in peroxisomes before it is routed to the ER and may become incorporated in phospholipids. This implies that peroxisomes contribute to the synthesis of lipids, which are in turn required for their own biogenesis/membrane plasticity [8].

The C-tail anchored peroxisomal membrane protein ACBD5, a member of the acyl-CoA binding domain protein family, is involved in the formation of peroxisome–ER contacts (see Section 3.3). ACBD5 interacts with the C-tail anchored ER-resident membrane protein VAPB (vesicle-associated membrane protein-associated protein B) to tether both organelles [52,53]. Protein interaction is mediated by the ACBD5 FFAT (two phenylalanines in an acidic tract)-like motif, which binds to the MSP (major sperm protein) domain of VAPB (Figure 2). It has been shown that the peroxisome–ER contact sites play a role in peroxisome positioning/motility, in cooperative metabolic processes (e.g., ether lipid synthesis) as well as in peroxisome membrane expansion [75]. Loss of ACBD5 or VAPB results in a shortening of the highly elongated peroxisomes in MFF-deficient cells, likely due to the interrupted membrane lipid transfer from the ER [52]. A transfer of ER lipids is also suggested by observations in MIRO1-expressing cells (see above). MIRO1/motor protein-mediated pulling forces generate membrane protrusions, which have a much higher surface area than the mother peroxisome they are generated from. It is likely that the mother peroxisome does not possess sufficient amounts of membrane lipids to allow for the generation of such elongated tubules [31]. How membrane lipids are transferred from the ER to peroxisomes is unclear. Although ACBD5 has acyl-CoA binding activity, it is suggested that this function is used to sequester and deliver very long-chain fatty acids (VLCFA) to the peroxisomal ABCD1 (ATP-binding cassette D1) transporter for uptake into peroxisomes and subsequent β-oxidation [76] (see Section 3.3). Specific lipid transfer proteins at peroxisome–ER contact sites may be involved in lipid transfer, such as oxysterol-binding proteins (ORPs), which can shuttle individual phospholipids. Recently, a role for VPS13D (vacuolar protein sorting-associated protein 13D) in peroxisome biogenesis was reported [77]. VPS13D can interact with MIRO1 and VAPB to form a bridge between the ER and peroxisomes [78]. VPS13D is a large protein with a hydrophobic groove, which could allow lipid channeling between organelles [79]. Such a “bulk flow” of lipids would be consistent with the observed rapid elongation processes of the peroxisomal membrane [31].

### 2.5. Multiple Roles of PO Membrane Dynamics

As outlined above (see Section 2), peroxisome membrane expansion results in membrane growth, which is linked to the multiplication/proliferation of peroxisomes by fission. However, the formation of peroxisomal membrane protrusions has also been linked to organelle interaction and communication. Interestingly, PEX11β was found to be co-regulated with proteins of the mitochondrial ATP synthase complex in a large-scale mapping approach, suggesting coordination of peroxisomal and mitochondrial functions [27]. Expression of PEX11β promotes peroxisome protrusions and is required for their formation [27,31]. These protrusions were observed to interact with mitochondria in mammalian cells and may facilitate metabolite exchange (e.g., for cooperative fatty acid β-oxidation and exchange of cofactors) and/or contribute to redox homeostasis [27]. Similar observations were made in plant cells, where light stress induced long peroxisomal membrane extensions (peroxules) which interacted with mitochondria [80,81]. The formation of those protrusions was dependent on plant *At*PEX11a [82]. As peroxisomes in mammalian cells are often in close contact with and tethered to the ER, such protrusions may support simultaneous interaction and communication with a third organelle, e.g., mitochondria.

## 3. Disorders of Peroxisome Dynamics and Plasticity

### 3.1. DRP1 Deficiency

DRP1 (approx. 80 kDa) (also known as DLP1; Dmn1 in yeast) is a highly conserved GTPase of the dynamin protein superfamily. Knockdown of DRP1, or overexpression of a dominant-negative mutant lacking GTPase activity, causes an elongated mitochondrial and peroxisomal phenotype in mammalian cells, consistent with a reduction in organelle fission and demonstrating the importance of DRP1 for mitochondrial and peroxisomal division [83,84,85]. It is primarily cytosolic, existing in this state as soluble dimers/tetramers, but is recruited to organelle membranes at sites of constriction by adaptors including MFF and FIS1 (see Section 2.2), forming higher-order oligomers to encircle the organelle in question [86]. DRP1 consists of an N-terminal GTPase domain, which hydrolyzes GTP to further constrict the membrane; a middle domain, mediating self-interaction; a less-conserved variable domain, which prevents cytosolic aggregation but promotes oligomerization upon membrane recruitment; and a GTPase effector domain (GED), which together with the middle domain mediates intramolecular interactions that stimulate GTPase activity [86,87] (Figure 2). DRP1′s ability to promote mitochondrial fission has been shown to be bidirectionally regulated by a number of post-translational modifications, including phosphorylation, SUMOylation, ubiquitination, S-nitrosylation and O-GlcNAcylation [88,89,90,91].

Patients with at least 20 distinct mutations in the *DMN1L* gene, encoding DRP1, have been described (OMIM: #614388, #610708) [92]. The majority are de novo heterozygous missense mutations (Appendix A), mostly in the GTPase and middle domains of the protein, and often with a dominant-negative effect (Figure 2). These mutations would be predicted to compromise both mitochondrial and peroxisome division since DRP1 is shared between the two organelles; however, the relative contribution and possible interplay of these two cellular defects to the pathology remains to be seen. Symptoms of patients with mutations in DRP1 are heterogeneous but predominantly neurological, typically including seizures, developmental delay, optic atrophy and hypotonia [92]. Neurological defects are common features of disorders with mitochondrial involvement, due to the high energetic demands of neurons [93]. Phenotypes consistent with peroxisomal dysfunction, such as hypomyelination, developmental abnormalities and nystagmus, are also sometimes present in patients with DRP1 mutations [94,95]. Analysis of cells derived from these patients normally show major perturbations to the mitochondrial network reflecting a fission defect, including elongation of mitochondria and formation of a hyperfused network [94,95,96,97,98,99,100,101], or swollen ‘balloon-like’ mitochondria [102,103,104], with abnormal cristae structure [102,105]. Where peroxisomal morphology has been investigated, an elongated phenotype is often seen [96,100,101,103,104], which may be accompanied by constriction [95] (Figure 3); however, in other patients, peroxisomes are morphologically normal [106,107]. Notably, in the unusual example of siblings with compound heterozygous frame shifts leading to no detectable DRP1 expression, the mitochondrial morphology in the brain was cell-type specific, with hippocampal and Purkinje neurons showing giant mitochondria, while the mitochondria in glia and non-neuronal cells appeared normal [102]. Therefore, it is possible that differences observed in peroxisome morphology between patients (and even between different studies on the same patient cells [105,107]), may also arise from differences in the cell type/cellular environment rather than the specific mutation.

Interestingly, despite the observed morphological changes, the metabolic functions of mitochondria and peroxisomes are only mildly affected (if at all) in patients with DRP1 mutations. Lactate levels may be elevated in some cases [95,96,107], presumably as a result of compromised mitochondrial respiration [98,105,106]; however, this feature is not shared by all patients [99,103]. Unlike patients with peroxisome biogenesis disorders (PBDs), those with DRP1 mutations do not typically show an increase in serum VLCFA levels [103,106,108], nor do their fibroblasts show defects in peroxisomal β-oxidation [96], indicating that the elongated peroxisomes can still perform their usual metabolic functions. How, therefore, the DRP1-dependent defects in mitochondrial and peroxisomal morphology lead to disease, without significantly altering cellular metabolism, remains an intriguing question (see also Section 3.2). One possibility is that, in the absence of the ability to divide, peroxisomes and mitochondria fail to dynamically adapt according to cellular needs. Since DRP1-dependent fission is also required for normal mitochondrial distribution within a cell [84], another contributing factor could be that, in the absence of DRP1 function, organelles are aberrantly dispersed through the cell, leading to local disruption of metabolism. Accordingly, several studies have reported an abnormal distribution of mitochondria and/or peroxisomes in patient fibroblasts [96,101,103] or cells expressing DRP1 harboring patient mutations [108].

Global *Dnm1l* knockout (KO) is embryonically lethal in mice (at ~E11.5) [109], which may explain why the vast majority of observed DRP1 patients have heterozygous missense mutations. Interestingly, *Dnm1l*^+/−^ heterozygous mice are viable and fertile with no gross morphological abnormalities, despite expressing only 25% of the wild type DRP1 protein. Global *Dnm1l* KO embryos are significantly smaller than wild type littermates, indicative of developmental delay, while cerebellum-specific KOs, which survive marginally longer until ~36 h after birth, show further defects in cerebellar development [109]. A contributing factor to the compromised neural development may be the reduction in developmentally regulated apoptosis seen in the neural tubes of *Dnm1l* KO embryos. Fibroblasts derived from global *Dnm1l* KO mice embryos (MEFs) show elongated, highly connected mitochondria with normal cristae, as well as elongated peroxisomes, as predicted for a defect in fission of both these organelles. This phenotype is also seen in KO HeLa cells, arising from a reduced number of fission events as determined by live cell imaging, and is reversible upon re-expression of DRP1 [85]. Interestingly, despite the morphological changes in *Dnm1l* KO MEFs, intracellular ATP production is unaffected [109], providing further evidence that changes in organelle dynamics can lead to pathology in the absence of significant metabolic disruption.

### 3.2. Mitochondrial Fission Factor (MFF) Deficiency

MFF (approx. 38 kDa) is a C-tail anchored membrane protein only found in metazoans, with a short C-terminal tail reaching into the organelle (mitochondrial inner membrane space; peroxisome matrix) and the N-terminus facing the cytosol (Figure 2) [61,110]. As such, it is oriented to be an ideal membrane adaptor for DRP1 recruitment to constriction sites (see Section 2). MFF contains two short repeat motifs and a coiled-coil domain, allowing oligomerization [111,112], as well as several phosphorylatable serine and threonine residues for regulation. MFF is a target of the energy-sensing adenosine monophosphate (AMP)-activated protein kinase (AMPK), with MFF phosphorylation promoting mitochondrial division [113]. Dynamic MFF oligomerization is required for DRP1 activation and division of mitochondria and peroxisomes [114]. MFF assembles in puncta on the organelle membranes, which also depends on its oligomerization. Interestingly, mutation of the coiled-coil domain appears to impact its peroxisomal localization [114]. MFF can promote mitochondrial and peroxisomal division independent of FIS1 [111,115,116,117] and can interact with PEX11β on peroxisomes [42,57] (Figure 1). Knockdown or knockout of MFF inhibits mitochondrial and peroxisomal division and results in the hyper-elongation of mitochondria and peroxisomes in multiple cell types, since DRP1 is no longer sufficiently recruited to the organelle membrane, indicating that MFF is the central adaptor for DRP1 at mitochondria and peroxisomes [57,61,110,115,117,118].

Several patients with MFF deficiency, a rare autosomal recessive neurological disorder (OMIM#617086), have been identified. This is caused by various mutations in the *MFF* gene, all of which (except one) lead to a truncated protein lacking the C-terminal TMD and tail (Appendix A) [c.C190T:p.Q64* [119]; c.184dup:p.L62Pfs*13 combined with c.C892T:p.R298* [120]; c.453_454del:p.E153Afs*5 [120]); c.C892T:p.R298* [121]; c.C433T:p.R145* [122]; c.19_20delAGinsTT:p.S7F [123]]. MFF truncations with a loss of the C-terminal TMD and tail will abolish the targeting and membrane localization of MFF, resulting in its absence at mitochondria and peroxisomes. Those patients show neurological abnormalities with onset during the first year of life and may present with Leigh-like encephalopathy, developmental delay, peripheral neuropathy, optic atrophy, and microcephaly [119,120,121,122]. A recent case with an amino acid change from serine to phenylalanine at position 7 was reported [123]. The 5-year-old male patient presented with cerebral palsy and encephalopathy, but without seizures or vision abnormality. Surprisingly, the mother, who is phenotypically normal, was determined to be homozygous for the same variant. Moreover, the last child delivered was also tested to be homozygous but is phenotypically normal (at age of 11 months). It is currently unclear, therefore, if this variant is of unknown significance (VOUS) or actually pathogenic.

Analysis of patient skin fibroblasts confirmed a loss of MFF function with mitochondrial and peroxisomal hyper-elongation [55,63,119,120,121] (Figure 3). Determination of peroxisomal biochemical parameters for fatty acid α- and β-oxidation, plasmalogen biosynthesis, or matrix protein import/processing did not reveal any deficiencies in these fibroblasts [55,120,121]. Plasma lactate in patients was only occasionally increased, but mitochondria in MFF-deficient patient fibroblasts showed no significant changes in oxidative phosphorylation or mtDNA [120,121]. Loss of MFF did also not significantly affect the mitochondrial membrane potential, ATP levels or the redox potential of the mitochondrial matrix in neuronal cells [124]. These findings indicate that defects in the membrane dynamics and division of mitochondria and peroxisomes rather than loss of metabolic functions contribute to the pathogeny of MFF deficiency.

Similar to DRP1 deficiency (see Section 3.1), it is currently unclear if and how alterations in both mitochondrial and peroxisomal dynamics contribute to the pathology of MFF deficiency. Besides bioenergetics defects, altered mitochondrial dynamics have been linked to neurodegenerative disorders [125]. A block in mitochondrial division, which results in larger organelles, may impact their transport along axons, organelle motility and positioning (e.g., at synaptic spines), and synaptic homeostasis [126,127,128]. Loss of DRP1 has been linked to defects in apoptosis in KO mice (see Section 3.1), which impacts brain development. It is likely that loss of MFF causes similar alterations, which may explain the developmental delay and neuronal abnormalities in patients. In addition, enlarged mitochondria may interfere with mitophagy, which involves engulfment of the mitochondrion with an autophagosomal membrane. This may also apply to peroxisomes, which are important for brain development and function [129]. However, recent studies with MFF-deficient skin fibroblasts have revealed that the hyper-elongated peroxisomes can still be degraded when autophagy/pexophagy is triggered [55]. Ultrastructural and immunofluorescence analyses revealed that the tubular peroxisomes in MFF-deficient fibroblasts originate from a spherical mother peroxisome, which is in contact with the ER [54]. This is likely the site of membrane lipid transfer from ER to peroxisomes. Loss of the peroxisome–ER tether proteins ACBD5 or VAPB in MFF-deficient cells leads to shorter peroxisomes, indicating that a reduction in peroxisome–ER contacts decreases the lipid transfer required for membrane elongation [52]. Electron-microscopy studies also revealed that the hyper-elongated peroxisomes in MFF-deficient fibroblasts are not constricted, which is in contrast to DRP1 deficiency (see Section 3.1) [47,55] (Figure 3). This indicates that a defect in MFF affects peroxisome division earlier than a defect in DRP1, and results in a defect in maturation. In line with this, only the mother peroxisomes, from which the membrane tubules emerge, appear to be import competent for peroxisomal enzymes. The tubules seem to represent pre-peroxisomal membrane structures, indicating that MFF-deficient fibroblasts accumulate peroxisomal membranes, but are reduced in the number of mature/fully functional peroxisomes. In addition, an unusual distribution of peroxisomal marker proteins has been observed [55], e.g., PEX14, a membrane component of the matrix protein import machinery, is enriched at tubular peroxisomes and may support interaction with microtubules to stabilize the tubular structures. Furthermore, the MFF-deficient cells display alterations in the peroxisomal redox state and intra-peroxisomal pH [55]. It is possible that these alterations contribute to the pathophysiology of MFF deficiency. The reduced number of mature, functional peroxisomes may also be less able to cope with environmental/metabolic changes, which require an increase in peroxisome numbers (e.g., an increase in dietary fatty acids, which require peroxisomal β-oxidation). This may explain the mild alterations of peroxisomal metabolism, which are occasionally detected in patients with defects in peroxisomal dynamics and division [96,130,131]. Furthermore, peroxisomes in MFF-deficient cells may be less able to cope with a stimulation of peroxisome proliferation and induced expression of peroxisomal matrix enzymes and/or membrane proteins [55]. Those proteins may accumulate in the cytoplasm, where they are either degraded or mistargeted to mitochondria, which may cause impairment of mitochondrial function [132,133].

*Mff* KO mice have been generated to model MFF deficiency [134]. *Mff* KO mice die at week 13 due to severe dilated cardiomyopathy, which results in heart failure, presumably caused by mitochondrial abnormalities. This is in contrast to the neurological symptoms diagnosed in MFF-deficient patients. Whereas mitochondria and peroxisomes in Mff-deficient MEFs are highly elongated, they do not show a substantial change in length in Mff-deficient mouse cardiomyocytes. Instead, differences in mitochondrial shape and abundance were observed [134]. This may indicate that morphological alterations of peroxisomes and mitochondria are cell-type specific (see Section 4) [55]. As morphological studies have mainly been performed with patient skin fibroblasts, we do not currently know much about peroxisomal and mitochondrial morphologies/dynamics in other cell types of MFF-deficient patients. However, as biochemical markers are not or only slightly altered in disorders affecting organelle dynamics, the morphological analysis of patient fibroblasts is a valuable diagnostic tool.

### 3.3. ACBD5 Deficiency

The acyl-CoA binding domain-containing protein 5 (ACBD5) (approx. 60 kDa) is a member of a family of seven human proteins, which can bind acyl-CoA and potentially other hydrophobic hydrocarbon compounds in order to ensure their solubility and stability in an aqueous environment. While ACBD1 and ACBD7 are small proteins consisting largely of the acyl-CoA binding domain (ACB) alone, the other members exhibit several different additional domains, which determine their individual function [135]. In particular, ACBD5 and its smaller orthologue, ACBD4, possess a C-terminal transmembrane domain, to anchor the protein in the peroxisomal membrane, while the ACB domain is closest to the N-terminus and faces the cytosol (Figure 2). In addition, both proteins contain the already described FFAT motif for the interaction with ER-resident VAP proteins, thereby facilitating membrane contact between the ER and peroxisomes [52,53,136] and several coiled-coil structure motifs, which might permit further protein interactions. According to this domain structure, ACBD5-associated disorders will likely develop a mosaic phenotype mirroring the combined loss of both protein functions: the binding of very-long chain (VLC) acyl-CoAs and the formation of peroxisome–ER membrane contact sites. Here, we will therefore discuss the pathological phenotype observed in human ACBD5-deficient patients and a corresponding mouse model in light of these two independent or combined ACBD5 functions.

The first two patients with an ACBD5 deficiency (OMIM: #618863) were identified in a screen for candidate disease genes in retinal dystrophy patients [137] (Appendix A). During the last five years, five further patients from four families with an ACBD5 deficiency have been reported in the literature [76,138,139,140], documenting the increasing alertness for this relatively newly described disorder (Appendix A). By contrast, no patients with a pathogenic mutation in the *ACBD4* gene have been detected to date. All ACBD5-deficient patients so far exhibit nonsense mutations in the *ACBD5* gene, which result in either premature truncation of translation or nonsense-mediated mRNA decay and therefore lead to a complete absence of the protein. The disease-related alterations observed in the patients point to a predominantly neurological pathology and include a visual dysfunction with nystagmus, progressive and eventually severe motor dysfunction with ataxia and dysarthria, which were reported for all patients, as well as cognitive decline, dysphagia [138], intentional tremor and seizures [139]. Correspondent with the neurological pathology, brain MRIs of the patients exhibited signs of hypomyelination in the deep white matter of the telencephalon, brain stem long fiber tracts and cerebellar peduncles [76,138,139]. Furthermore, in the oldest patient reported so far (age 36 years), atrophic alterations, such as diffuse ventricular enlargement, widening of the cerebral sulci and space between cerebellar folia were observed [138]. Concerning the visual impairment, all patients from the literature exhibit a severe cone-rod dystrophy. Moreover, an additional optic nerve pallor indicating a degeneration of ganglion cells was detected in the 36-year-old patient, which might suggest a more advanced stage of the disorder. At the biochemical level, the patients exhibit only moderately elevated plasma levels of the VLCFA hexacosanoic acid (C26:0) if compared to, for example, the average of X-ALD patients: ACBD5-deficient patients exhibit an average elevation of 1.2-fold over average values obtained for healthy individuals, compared to an average of 4.5-fold for X-ALD patients [141]. Of note, all parameters for other peroxisomal metabolites in blood plasma were not significantly altered if compared to healthy individuals. In line with these observations, lipidomics of ACBD5-deficient patient fibroblasts revealed an accumulation of VLCFA in most phospholipid classes [142,143]. All these data point to a function of ACBD5 in the pathway of peroxisomal VLCFA β-oxidation, potentially as an acceptor for a broad spectrum of VLC-acyl CoAs to be imported into peroxisomes [143]. Unexpectedly, a decrease in ether phospholipids in cultured cells suggests that ACBD5 function is not restricted to its supposed role in peroxisomal β-oxidation, since peroxisomal ether lipid synthesis does not require enzymes from the β-oxidation pathway.

Recently, an *Acbd5* KO mouse model was used to further link cellular and biochemical analyses with the organ-specific pathology of ACBD5 deficiency [144]. Like human patients, the mice show moderate increases in VLCFA in blood plasma and tissues and develop a neurologic pathology characterized by a progressive motor dysfunction and retinal degeneration. Among other neurological symptoms, a very prominent feature of the mouse phenotype is their severe spinal kyphosis, which is, however, not accompanied by major changes in bone morphology (Figure 4). Hence, this phenotype is rather the result of a neuromuscular pathology. Such misbalances in trunk muscle coordination as well as the intentional tremor and nystagmus observed in human patients point to a cerebellar contribution to the disease pathology.

In line with this, ACBD5-deficient mice show signs of neuronal degeneration, such as axon swellings and moderate hypomyelination of cerebellar Purkinje cells, which lead to significant decline in Purkinje cell number in one-year-old mice [144]. Moreover, the activation of astroglia and microglia imply that inflammatory processes in the affected neural tissue accompany the neuronal degeneration. To link lipid changes with the disease pathology, tissue samples from cerebellum and liver, which exhibit no obvious pathologic alterations at the organ level, were compared by lipidomics. As expected, phospholipids contained increased amounts of VLCFA in both tissues [144]. Remarkably, the cerebellum contained phosphatidylcholine (PC) species with increased amounts of polyunsaturated fatty acids (PUFA) with a very high chain-length (up to C38). Such a profound increase in fatty acid chain length was not observed for cerebellar phosphatidylethanolamines (PE) nor for any lipid class from liver. These ultra-long chain fatty acids (ULCFA) can only be synthesized by the elongase of very long chain fatty acids (ELOVL) ELOVL4, which shows highest expression in the retina and cerebellum but is not found in the liver [145]. On the contrary, ether phospholipid classes (PC[O], PE[O]) were in general decreased in the cerebellum but not in the liver and did not exhibit the shift in chain length observed for the PC. These observations underline the importance of a highly defined phospholipid environment in specialized tissues. Changes in a subset of lipid species, such as the ULCFA-containing PC of the cerebellum, may contribute to the tissue-specific pathology observed in peroxisomal disorders.

Since ACBD5 plays, in addition to its acyl-CoA binding function, a unique role as a tethering protein connecting peroxisomes with the ER, it is tempting to speculate how aspects of the pathology of the ACBD5 deficiency might originate from the reduction in peroxisome–ER contact zones. Such organelle contacts between the ER and peroxisomes have been supposed to be involved in the transfer of intermediates in pathways which are shared by both organelles, e.g., the synthesis of ether phospholipids or the PUFA DHA [146]. Measurements of DHA and ether lipid content in the membrane of ACBD5-deficient human cells revealed inconsistent data, reporting either reductions [53,142] or no major changes [143] for both parameters. Comparison of ether lipids in the liver and cerebellum of ACBD5-deficient mice also revealed alterations that are more complex: while ether phospholipid content in the cerebellum was reduced in ACBD5-deficient mice, no significant changes were found in the liver. Unexpectedly, however, alkyl-diacylglycerides were highly increased in the liver, suggesting that the loss in ER membrane contacts induces a tissue-specific compensatory response in the peroxisomal part of the pathway [144]. With respect to DHA, no significant changes could be detected in lysophosphatidylcholine (LPC) C22:6 species in either the liver or cerebellum. Additionally, by connecting ER and peroxisome membranes at contact sites, pathways of fatty acid elongation (ER) and degradation of VLCFA (peroxisomes) become linked as well. Hence, the physical proximity of both pathways may create a reciprocal loop to avoid excess VLCFA incorporation into membrane phospholipids (also being catalyzed at the ER). Indeed, fatty acid spectra in PC and LPC from ACBD5-deficient mice exhibit a shift toward longer and more saturated fatty acids in the liver and highly elongated PUFA in the cerebellum, which coincides with the activities of ELOVL1 and ELOVL4 in the respective tissue. While ELOVL1, which elongates saturated fatty acids above C22, is ubiquitously expressed [147], ELOVL4 is preferentially expressed in brain and testis and accepts both saturated as well as unsaturated C28-C38 fatty acids as a substrate [148]. Thus, the tissue-specific fatty acid changes in phospholipid composition may reflect an uncontrolled fatty acid elongation as a result of the loss of counteracting peroxisomal β-oxidation at peroxisome–ER membrane contact sites.

In addition to their potential significance for specific lipid pathways, peroxisome–ER contact sites appear to be involved in the general transfer of phospholipids from the ER, as the site of synthesis, to peroxisomes in order to provide material for membrane expansion in the process of peroxisome growth and division (see Section 2). As previously discussed, silencing of ACBD5 or VAPB in MFF- or DRP1-deficient cells results in a decrease in peroxisome tubule length, implying a decrease in the membrane lipid transfer required for tubule elongation [52,53]. To substantiate the findings from division-defective mutant cells, peroxisome elongation was induced by treatment with DHA in ACBD5-deficient and wild type MEFs [144]. Confirming a reduced capacity for membrane expansion under these more physiological conditions, peroxisomes from ACBD5-deficient MEFs showed a refractory response to the DHA exposure, in contrast to wild type controls. However, it should be noted that none of the ACBD5-deficient cell types investigated so far, which all exhibit significant reductions in peroxisome–ER membrane contacts, exhibit a reduced peroxisome abundance. Hence, any reduction in peroxisome growth and division as a result of a disrupted transfer of membrane lipids from the ER to peroxisomes has to be compensated for, for example, by lipid transfer through the remaining peroxisome–ER contacts. Future studies are required to clarify, whether peroxisome maintenance is fundamentally remodeled in ACBD5-deficient cells in order to cope with the quantitative reduction in peroxisome–ER contact site numbers.

In polarized cells, such as neurons, kidney epithelial cells or retinal pigment epithelium, peroxisomes are not evenly distributed across the cellular interior, but accumulate at distinct cellular locations [149,150,151]. To guarantee peroxisome positioning, ER contacts might be used to tether peroxisomes at distinct locations in order to avoid active transport by microtubules. As described above, silencing of ACBD5 in fibroblasts and HeLa cells increased peroxisomal motility [52,53], whereas ACBD5 overexpression decreased long-distance movements in hippocampal primary neurons [152]. In the polarized neurons, the decreased motility in response to ACBD5 overexpression was accompanied by a peroxisome relocation from the dendritic compartment to the neuronal soma, implying that an imbalance between microtubule transport and ER anchoring might affect organellar positioning. Ultrastuctural observations from ACBD5-deficient mouse hepatocytes also suggest that the loss of ACBD5 alters intracellular peroxisomal distribution [144]. However, hepatocytes lacking ACBD5 showed in addition to a general increase in peroxisome numbers, focal accumulations of tightly associated/clustered peroxisomes. Such a distribution pattern cannot be explained by the enhanced motility of the organelles as result of the disrupted ER membrane contacts. Rather, uncontrolled peroxisome proliferation accompanied by an imbalance in the hepatocyte organelle interaction network seems to induce this cytomorphological phenotype as a result of the lost peroxisome–ER interaction, which in hepatocytes often appear as so-called wrappER contacts with the rough ER [153]. Future studies have to show whether these changes in organelle localization have an impact on the metabolic alteration observed in mice and humans.

In summary, the phenotype of ACBD5 deficiency, with its peculiar focus on the degeneration of the central nervous system (including retina), appears to be based on a unique pathological mechanism, which might combine alterations induced by the loss in acyl-CoA binding and peroxisome–ER tethering. It will be challenging to unravel, in the future, to what extent the disruption of the protein’s unique tethering function contributes to the disease pathology, either by altering intrinsic cellular phospholipid flow required for peroxisome maintenance/dynamics, shifting peroxisome distribution and motility within the cell, or by dysregulation of metabolic networks of cellular lipid metabolism.

### 3.4. PEX11β Deficiency

Mammals express three isoforms of the peroxisomal protein PEX11: α, β and γ. As discussed in Section 2, PEX11β (28 kDa) is the major regulator of peroxisome growth and division, being required for the deformation and elongation of the membrane as well as the assembly of the fission machinery [8] (Figure 1), while the roles for PEX11α and PEX11γ are less clear. Unlike the other fission factors discussed here, under physiological conditions, PEX11β is only found at peroxisomes and not mitochondria [48]. In addition to proliferation, the single yeast PEX11 homologue has been implicated in other peroxisomal functions, including mediating membrane contact sites with the ER [154] and forming pores in the membrane [155], though whether these are conserved in mammals is disputed [133]. As outlined in Section 2.1, PEX11β is a membrane-spanning protein with the N- and C-termini facing the cytosol (Figure 2). Whereas the N-terminus with its amphipathic helices is required for lipid interaction, membrane elongation and the oligomerization of PEX11β [41,43,45,49], the conserved short C-terminus and glycine-rich internal loop appear to be important for the interaction with FIS1 and promotion of peroxisome division [46,63].

To date, patients across five families have been identified to have mutations in the *PEX11B* gene (Appendix A), leading to a disease categorized as peroxisomal biogenesis disorder 14B (OMIM: #614920) [130,131,156]. In all but one of these cases (a compound heterozygous nonsense mutation and deletion [131]), these are homozygous nonsense mutations leading to a premature stop codon and, therefore, a truncated protein—where determined, this leads to no detectable expression of PEX11β [130]. Patients with mutation in *PEX11A* or *PEX11G* have yet to be described. The clinical profile of PEX11β-deficient patients matches a mild PBD, with patients presenting with mild intellectual disability, congenital cataracts, hearing defects and short stature. However, unlike PBDs, but similar to the other defects in organelle dynamics described here, most of the PEX11β-deficient patients described show no alterations in peroxisomal metabolism, including levels of VLCFA and plasmalogens in the blood, suggesting no major effect on peroxisome metabolic function [130,131,156]. Fibroblasts from one PEX11β-deficient patient have been characterized and show a reduced number of elongated and enlarged peroxisomes but normal peroxisomal α- and β-oxidation [130] (Figure 3). This peroxisomal morphology points to a defect in fission, while elongation can still occur to some extent, suggesting another PEX11 isoform may partly compensate for the role of PEX11β in membrane expansion. In most cells, in contrast to Zellweger syndrome PBDs (ZS), these aberrant peroxisomes are import competent; however, in ~10% of cells, the matrix protein catalase is not imported and is present in the cytosol instead [130]. Furthermore, PEX14 is mistargeted to mitochondria, indicating that the reduction in peroxisome numbers results in an accumulation of nascent peroxisomal proteins in the cytosol (Figure 3). Interestingly, culturing these patient-derived fibroblasts at 40 °C exacerbates the peroxisomal dysfunction, with accumulation of VLCFA and failure to import catalase in ~90% of cells. Of note, PEX11γ mRNA and protein levels in fibroblasts are strongly reduced at 40 °C, which may explain this phenotype if PEX11γ is partly compensating for the loss of the PEX11β function. Consistent with this, overexpression of PEX11γ, but not PEX11α, can partially rescue the catalase import defect and altered peroxisome morphology in patient fibroblasts under these conditions, though not to the extent of re-expressing PEX11β [130]. This indicates that a better understanding of the roles and regulation of the other PEX11 isoforms, particularly PEX11γ, may be therapeutically useful for treating PEX11β deficiency in the future [157].

Much like PEX11β-deficient patients, homozygous *Pex11**b* KO mice display some pathological symptoms reminiscent of PBDs, but do not exhibit the same cellular or metabolic phenotypes [158]. *Pex11b*^+/−^ heterozygotes show no macroscopic phenotype, but on a cellular level typically show an intermediate phenotype between wild type and homozygous littermates [158,159]. Global KO of PEX11β in mice seems to result in a more severe phenotype than in human patients, leading to death shortly after birth. *Pex11**b* KO pups show developmental defects at P0.5 relative to wild type littermates, including being undersize and underweight, having underdeveloped liver and kidneys and a delay in skull ossification. Both embryonic and neonatal *Pex11**b* KO mice display neurological abnormalities also seen in ZS models, including defects in neuronal migration and an increase in the apoptosis of neocortical neurons [158], which may underpin the patients’ neurological symptoms. Neuronal cultures from *Pex11**b* KO embryos show a decrease in both networking/branching and expression of mature/synaptic neuronal markers relative to wild type, as well as an increased production of ROS, which could also contribute to the observed neurological phenotypes [159]. *Pex11a* KO mice, on the other hand, are viable with no obvious pathological phenotypes under normal conditions [160], though they do develop non-alcoholic fatty liver disease when fed a high-fat diet [161]. Basal peroxisome abundance and morphology are unaltered in *Pex11a* KO MEFs and liver [160], with most changes being seen in peroxisome metabolism rather than the dynamics [162,163]. This, together with the inability of PEX11α to compensate for PEX11β deficiency in patient fibroblasts, supports the notion that the α and β isoforms have distinct functions, with PEX11α likely having a more prominent role in the metabolic functions of peroxisomes.

In contrast to ZS, and consistent with the results obtained from PEX11β-deficient fibroblasts, both PTS1- and PTS2-dependent matrix protein import is normal in *Pex11**b* KO MEFs. Supporting a peroxisome-specific fission defect, peroxisomes are observed to be elongated, clustered and reduced in number by electron-microscopy in hepatocytes from *Pex11**b* KO mice, with mitochondria displaying a normal morphology and, interestingly, possibly showing some proliferation relative to the wild type, presumably as a compensatory mechanism [158]. Despite the ZS-like symptoms, biochemical peroxisome parameters are normal or only slightly affected in *Pex11**b* KO mice, in a tissue-specific manner—in the brain, VLCFA levels are normal while plasmalogens are mildly reduced, whereas in the liver plasmalogen levels are normal while VLCFA slightly accumulate. Similarly, measures of enzymatic activity in fibroblasts show normal plasmalogen synthesis and phytanic/pristanic acid oxidation, and a small decrease in VLCFA β-oxidation [158]. Any alterations are of a much smaller magnitude than those seen in ZS patients/models, recapitulating the predominantly unaffected peroxisome function seen in the PEX11β-deficient patients. The shared symptoms between ZS and PEX11β-deficient patients (albeit being less severe in PEX11β deficiency) thus suggest that, at least in part, these aspects of the pathologies may not be a result of VLCFA toxicity or generalized loss of peroxisome metabolic functions. This further supports the notion that loss of organelle dynamics can lead to pathology independent of organelle metabolism. This is exemplified in certain inherited forms of noise-induced hearing loss, which are a result of defective peroxisome dynamics rather than metabolism. Under normal conditions, noise overexposure triggers the production of ROS in auditory hair cells of the inner ear, and this oxidative stress leads to peroxisome proliferation as an antioxidant defense. In patients where peroxisome plasticity is compromised, peroxisomes cannot be sufficiently upregulated in response to excessive ROS generation, so are unable to restore redox homeostasis, resulting in damage to the auditory hair cells and, thus, hearing defects [164]. Since peroxisome dynamics are defective in the absence of PEX11β, this mechanism could underlie the hearing problems experienced by PEX11β-deficient patients [130,131].

## 4. Modelling and Prediction of Peroxisomal Dynamics

As discussed, peroxisomes are highly dynamic and plastic organelles, which can adapt their number and morphology to changes in the environment (see Section 1) (reviewed in [39,165,166]). Those dynamic changes are often linked to a stimulation of peroxisome proliferation by peroxisome proliferators (e.g., hypolipidemic drugs, plasticizers) or fatty acids/lipids, which in mammals act via the peroxisome proliferator-activated receptor (PPAR) signaling pathway (reviewed in [8,167,168,169]), or during liver regeneration [170]. Differences in peroxisome morphology have also been observed in different tissues and cell types [68,171,172,173,174,175]. Similar peroxisomal morphologies and dynamic alterations were observed in mammalian cell culture. Elongation of peroxisomes was stimulated by a variety of experimental and culture conditions, including cell density, serum components/growth factors, fatty acids, microtubule depolymerization, UV irradiation [68,176,177], expression of PEX11β [48], or manipulation of the division machinery (e.g., of DRP1 or MFF) (see Section 3). Although PEX11β appears to play a key role in peroxisomal growth and division (see Section 2 and Section 3.4), little is known about the signaling pathways and gene responses regulating these processes in humans [157].

To explain the variation in peroxisome morphology and to make predictions with respect to peroxisome dynamics in health and disease conditions, we developed a simple mathematical model based on a stochastic, population-based modeling approach [31,55]. Briefly, peroxisome shape is determined by (i) membrane lipid flow into the peroxisome body (e.g., from the ER), (ii) elongation growth, and (iii) peroxisome division rate (Figure 5). Peroxisome turnover was also considered, and peroxisome parameters were based on experimental data, where possible. The morphological alterations of peroxisomes in MFF-deficient fibroblasts could be captured by simply reducing only one parameter, the division rate, to almost zero. As the membrane lipid flow rate and elongation growth speed remain unchanged, this results in a reduction in peroxisome numbers and a pronounced membrane elongation, as observed in MFF deficiency. The observation that healthy control fibroblasts usually display small, spherical peroxisomes, but form highly elongated peroxisomes when division is blocked, suggests that the membrane lipid flow rate, elongation growth speed and division rate are high in fibroblasts under normal conditions. This explains why elongated peroxisomes are rarely observed in normal fibroblasts. On the other hand, cell lines/cell types with a lower peroxisome division rate would display more elongated peroxisomes, e.g., as observed in HepG2 cells [68,157]. Cells with a low membrane lipid flow rate or elongation speed, but high division rate, would predominantly display small peroxisomes and reduced numbers. Such a scenario can be experimentally generated by the depletion of the peroxisome–ER tether ACBD5, which reduces peroxisome–ER tethering and membrane expansion [52]. The model may also explain why peroxisomes elongate after microtubule depolymerization in some cell types. The division rate may be promoted by pulling forces via MIRO/motor proteins along microtubules (see Section 2.4). Loss of microtubules may reduce (but not abolish) the division rate, which would in turn result in peroxisome elongation as observed in HepG2 and COS7 cells, likely mediated by PEX11β [68,69]. By altering the different parameters to recapitulate the peroxisomal phenotype seen in different pathologies, the model is also applicable to other disease conditions, e.g., the loss of PEX5 in Zellweger spectrum disorders [31].

The model parameters (i–iii) are likely to change during cellular growth and differentiation, presumably via cellular signaling and expression levels/post-translational modifications of tethering, membrane shaping and division components. We recently demonstrated that the interaction of the peroxisomal tether ACBD5 with ER-resident VAPB is regulated by phosphorylation [178], with phosphorylation in the core region of the ACBD5 FFAT motif inhibiting interaction with the MSP domain of VAPB. Overall, we suggest that peroxisome morphology is likely to be differently affected in various cell types of patients suffering from organelle division defects. Environmental changes (e.g., in diet or stress conditions) and associated signaling pathways that promote peroxisomal growth and division can potentially trigger the hyper-elongation of peroxisomes in formerly unaffected cell types and contribute to the pathophysiology of those disorders. It is therefore important to understand how organelle dynamics and peroxisomal growth and division are regulated in humans and what intracellular signaling pathways are involved.

## 5. Potential Strategies for Treatment

Like many inherited disorders, there are currently no clinically approved cures for patients with a defect in mitochondria and/or peroxisomal fission, and treatments are generally focused on alleviating symptoms, for example anticonvulsants to counter seizures [179]. Several MFF-deficient patients [121,122] and a PEX11β-deficient patient [130] have been reported to respond positively to treatment with a mitochondrial drug cocktail, which either offers mild improvement or at least halts deterioration. Such cocktails typically consist of antioxidants, such as riboflavin, thiamine, carnitine and Co-enzyme Q10, which also bolster mitochondrial function in patients where this might be compromised, e.g., by promoting electron transport chain function (Co-enzyme Q10, riboflavin) or mitochondrial β-oxidation (carnitine) [180]. The broad-spectrum PPAR agonist bezafibrate has shown promise in fibroblasts from a patient with a DRP1 (p.G362S) mutation, restoring ATP production to control levels and reducing mitochondrial elongation [181]. Since PPAR activation upregulates mitochondrial and, to a lesser extent in humans, peroxisomal biogenesis [157,182], this suggests that enhancing the dynamics of these organelles may be a useful therapy for patients with defects in the fission proteins. Beyond small-molecule modulators, gene augmentation therapy to reintroduce wild type copies of a mutated protein into particularly affected tissues may be a possible therapeutic avenue in the future, assuming that the dominant-negative effect of some of the mutant fission proteins can be overcome. Proof-of-principle gene therapy in a mouse model of PEX1(p.G844D) ZS has shown the feasibility of this approach in treating some symptoms of peroxisomal disorders, with AAV (adeno-associated virus)-mediated expression of human wild type PEX1 through sub-retinal injection preventing deterioration of peroxisome metabolism and cell function in the retina, if administered at early enough time points [183].

Recently, we reported that the peroxisomal fission defect in MFF-deficient patient fibroblasts can be rescued by overexpression of PEX11β, and vice versa [63]. This suggests there are two parallel pathways of peroxisome fission that can act independently—one requiring PEX11β (and, from our data, FIS1), and one requiring MFF—though the relative contribution of each under physiological conditions remains to be seen. Importantly, this demonstrates that the elongated peroxisomal phenotype in MFF-deficient and PEX11β-deficient patient fibroblasts can be reversed by the upregulation of the division pathway that is still intact. We anticipate, therefore, that pharmacological upregulation of MFF in PEX11β-deficient patients, and PEX11β in MFF-deficient patients, may be a viable therapeutic approach to restoring healthy peroxisome dynamics and thus treating the symptoms of peroxisomal dysfunction in these patients. However, a better understanding of how PEX11β and MFF expression are endogenously regulated would be necessary to pursue this. In this study we also showed that, in fibroblasts from a PEX19-deficient ZS patient (where peroxisome membranes are absent), overexpressed PEX11β instead inserts into the mitochondrial membrane where it can drive mitochondrial fission, presumably due to the dual localization of its division complex cofactors DRP1, MFF and FIS1 at both peroxisomes and mitochondria [10,11,25]. This suggests, depending on the level of endogenous PEX11β expressed in different tissues, that excess mitochondrial fission could also be a contributing factor to ZS pathology. Indeed, it has been proposed that the mislocalization of peroxisomal proteins to mitochondria in the absence of peroxisomes may underlie the mitochondrial dysfunction observed in ZS patients, and removing them from the mitochondrial membrane could be of therapeutic benefit [132].

Interestingly, aside from the inherited disorders with a genetic loss or mutation discussed above, dysregulated organelle dynamics are observed in a wide variety of pathologies, including neurodegeneration, cardiovascular disease and cancer [184,185,186], though it is not yet necessarily clear whether this is a cause or a consequence of the disease state. Usually, such diseases are associated with excess or pathophysiological fission, compromising organelle quality control in the cell. *FIS1* is amplified in certain cancers [184]; DRP1 and FIS1 levels are increased in post-mortem brain tissue from Alzheimer’s disease (AD), according to some reports [187]; the DRP1–FIS1 interaction is enhanced in amyloid-beta treated cells (modeling AD) and AD patient-derived fibroblasts [188]. As a result, compounds that inhibit fission have been suggested as therapies for a range of diseases. Mdivi-1, a small molecule considered (though controversially [189]) to act as a DRP1 inhibitor, has been shown to restore healthy mitochondrial morphology and improve cognitive decline in a mouse model of AD [190] and reduce neurotoxicity in Parkinson’s disease mouse models [191]. Similarly, the peptide P110, which blocks the DRP1–FIS1 interaction and therefore reduces DRP1 recruitment to mitochondria, rescued mitochondrial morphology and function in AD patient-derived fibroblasts and attenuated cognitive decline in an AD mouse model [188], and also reduced mitochondrial dysfunction and improved cardiac function in rats with myocardial infarction-induced heart failure [192]. Typically, only effects on mitochondria have been investigated, but since DRP1 and FIS1 also participate in peroxisome division, it might be expected that peroxisome dynamics are also dysregulated in these diseases and can be rescued using these compounds. It remains to be seen to what extent excess peroxisomal fission might contribute to these pathologies, and whether the inhibition of peroxisome division plays a role in the protective effects of these putative treatments. Regardless, it seems like the dysregulation of organelle division processes may be a common feature seen in a wide variety of pathological conditions, whether directly or indirectly. Indeed, an increase in DRP1 and FIS1 mRNA but a decrease in their protein levels in peripheral blood mononuclear cells isolated from blood samples is reproducibly seen in patients with the autoimmune disorder myasthenia gravis, prompting the proposal to use these as biomarkers for this disease [193]. Altogether, this raises the possibility that modulating organelle fission may be a future broad-spectrum treatment to improve cell performance and organelle function in a host of conditions.

## Figures and Tables

**Figure 1 cells-11-01922-f001:**
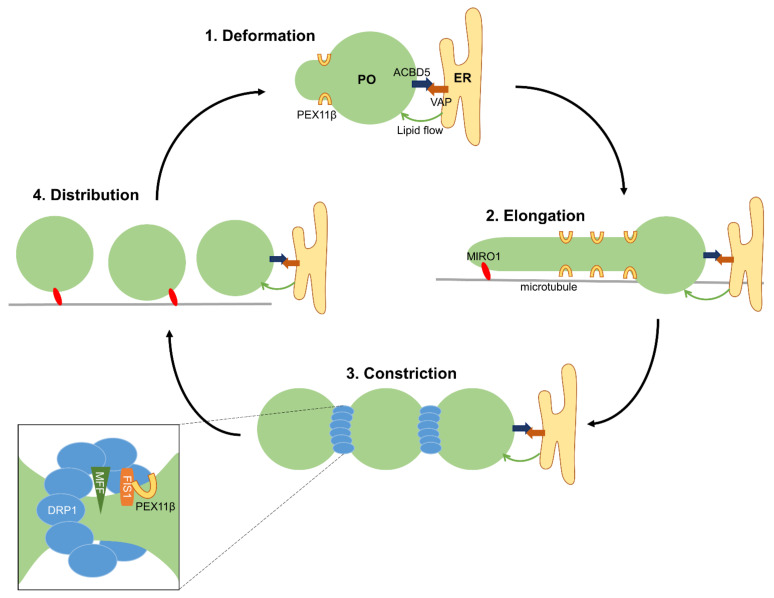
Growth and division of mammalian peroxisomes. Schematic of peroxisome proliferation from pre-existing peroxisomes, via the growth and division cycle. (1) The peroxisomal membrane undergoes initial deformation, requiring the N-terminus of PEX11β to disrupt the lipid bilayer. Extension of this protrusion is supported by lipid flow from the ER at ACBD5-VAP-mediated membrane contacts, via an unknown mechanism potentially involving lipid transfer proteins. (2) Elongation of the protrusion also requires PEX11β, aided by pulling forces from the movement of the peroxisomal MIRO1-motor protein complex along microtubules. (3) The nascent tubule undergoes constriction, allowing oligomerization of the GTPase DRP1 and leading to a characteristic ‘beads-on-a-string’ morphology. DRP1 is recruited to the membrane by binding to its adaptors FIS1 and MFF, which also interact with PEX11β (see cut-out). PEX11β facilitates DRP1-dependent GTP hydrolysis to drive further constriction and ultimately membrane scission, dividing the tubule into multiple ‘daughter’ peroxisomes. (4) These newly formed peroxisomes import new matrix and membrane proteins to become fully functional, mature organelles, and are distributed throughout the cell along microtubules by the MIRO-motor protein complex.

**Figure 2 cells-11-01922-f002:**
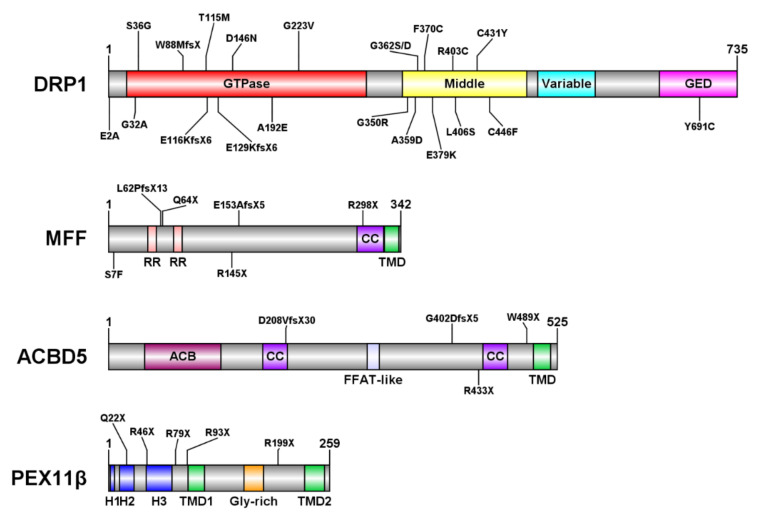
Domain structure of human DRP1, MFF, ACBD5, PEX11β and known pathogenic mutations. ACB, Acyl-CoA binding domain; CC, coiled-coil domain; FFAT, two phenylalanines in an acidic tract; GED, GTPase effector domain; Gly rich, glycine-rich region; H, helix; RR, repeat motifs; TMD, transmembrane domain. For details, see text. For information about patients, see Appendix A.

**Figure 3 cells-11-01922-f003:**
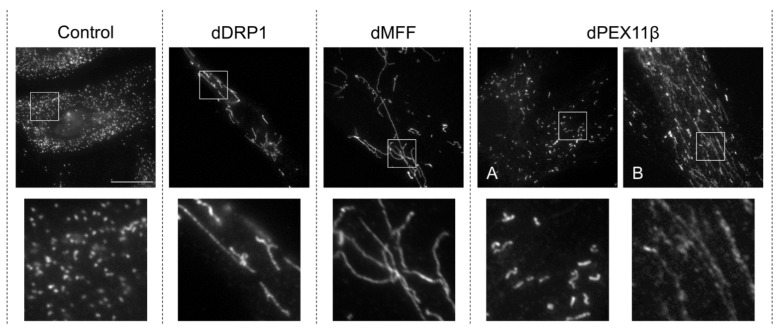
Peroxisome morphology in DRP1, MFF, and PEX11β-deficient patient fibroblasts. Cultured patient skin fibroblasts were processed for immunofluorescence and stained with antibodies to PEX14, a peroxisomal membrane marker. Note the hyper-elongated peroxisomes in DRP1 and MFF-deficient cells (dDRP1, dMFF). In contrast to dMFF cells, peroxisomes in dDRP1 cells are able to constrict. Peroxisomes in PEX11β-deficient fibroblasts (dPEX11β) are mainly rod-shaped (A). Note that PEX14 is often mistargeted to mitochondria in dPEX11β cells (B). Peroxisomes in ACBD5-deficient patient fibroblasts (not shown) are indistinguishable from controls. Higher magnifications of boxed regions are shown. Scale bar, 20 µm.

**Figure 4 cells-11-01922-f004:**
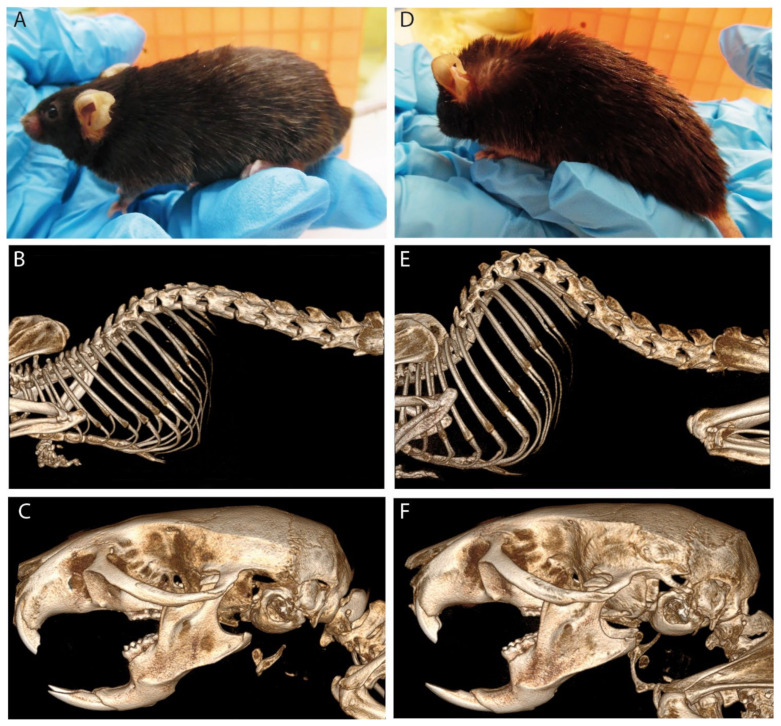
Phenotype of an ACBD5-deficient mouse line. (**A**) General body morphology of a 1-year-old wild type mouse compared to (**D**) an ACBD5-deficient mouse, which exhibits a prominent kyphosis in the thoracolumbar spine. Computer tomography (CT) analysis of the bone morphology revealed no obvious skeletal malformations in vertebrae of ACBD5-deficient mice (**E**) compared to wild type mice (**B**). Likewise, other bone structures (e.g., bones of the skull) show a comparable morphology in wild type (**C**) and ACBD5-deficient mice (**F**). Hence, the pathological kyphosis is not caused by compromised bone development but is of a secondary nature. As the animals also show a motor phenotype and degeneration of cerebellar Purkinje cells, the kyphosis is most likely caused by a compromised spinocerebellum inducing an imbalance in the contraction of flexor and extensor trunk musculature.

**Figure 5 cells-11-01922-f005:**
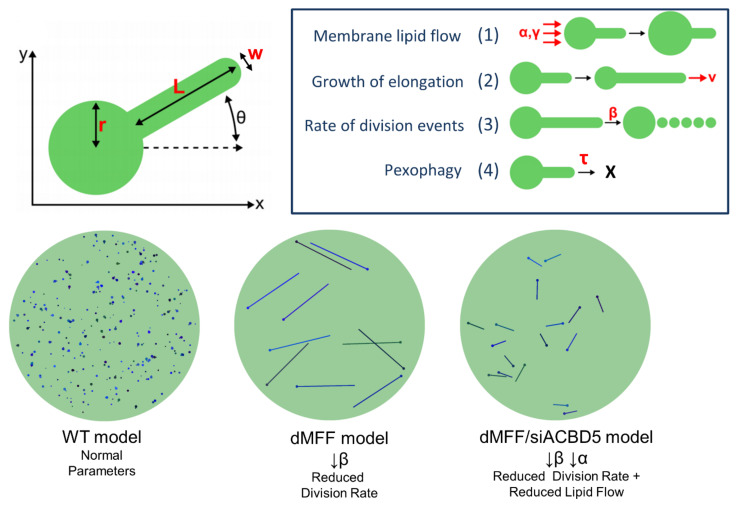
Mathematical model of peroxisome morphology and dynamics. Each peroxisome is represented as a spherical body of radius r and a cylindrical elongation of length L and diameter w. The model considers: (1) membrane lipid flow into the body (e.g., from the ER) (governed by rate α and lipid flow constant γ), (2) growth of the elongation (governed by speed v), (3) peroxisome division (with rate per unit length β), and (4) peroxisome turnover (“pexophagy”) (governed by the peroxisome mean lifetime τ). Snapshot of the model simulation for wild type cells (normal parameters), MFF-deficient cells (dMFF) (reduced division rate) and dMFF cells with reduced lipid flow to simulate silencing of ACBD5 (adapted from [29,50]).

## Data Availability

All datasets generated for this study are included in the article.

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
