# Peer review of "Fission Impossible (?)—New Insights into Disorders of Peroxisome Dynamics"

_cells, 2022, doi:10.3390/cells11121922_

Round 1

Reviewer 1 Report

The manuscript entitled Fission Impossible (?) – New Insights into Disorders of peroxisome Dynamicsby Ruth E. Carmichael, Markus Islinger and Michael Schrader, is comprehensive a review paper that provides a broad overview of structural and functional dynamics of peroxisome. The manuscript is well written and contains not very new (considerable amount of review article from the same authors) but valuable data.

However, authors should include some recent data regarding the origin of peroxisomes from mitochondria and the role of DRP1 in this (Sugiura A et al. doi: 10.1038/nature21375 and Aleksic M et al. doi: 10.3390/cells10092248).

Author Response

Comments and Suggestions for Authors:

The manuscript entitled “Fission Impossible (?) – New Insights into Disorders of peroxisome Dynamics” by Ruth E. Carmichael, Markus Islinger and Michael Schrader, is comprehensive a review paper that provides a broad overview of structural and functional dynamics of peroxisome. The manuscript is well written and contains not very new (considerable amount of review article from the same authors) but valuable data.

However, authors should include some recent data regarding the origin of peroxisomes from mitochondria and the role of DRP1 in this (Sugiura A et al. doi: 10.1038/nature21375 and Aleksic M et al. doi: 10.3390/cells10092248).

We would like to thank the reviewer for the positive evaluation of our manuscript. Besides a timely update on mutations/patients with defects in genes/proteins involved in peroxisome dynamics, the manuscript contains new data and summarises the findings of the last few years, which have not yet been reviewed, particularly not with a patient-centred focus.

As requested by the reviewer, we have added relevant data regarding the de novo formation of peroxisomes and the role of DRP1 in this (page 2, lines 63-66 in revised manuscript).

Reviewer 2 Report

In this paper, Carmichael RE and colleagues provide an extensive and very complete review on peroxisomal membrane dynamics particularly focusing on peroxisome biogenesis according to the growth and division model. The role of different division associated proteins in these processes, as well as the role of peroxisome-ER contacts in lipid transfer for peroxisomal membrane proliferation, is fairly discussed, and additionally, the authors provide a sum up of all the published data regarding clinical phenotypes and pathophysiology of patients with defects in these proteins and raise potential ways for the treatment of these disorders. I think the review is very well written and would be of great interest to readers of the journal and therefore I recommend its publication after some minor corrections.

General comments:

1. Some references are missing, while some are redundant. Authors should be careful when choosing the papers to cite. When review papers are needed to support a sentence, they tend to cite only their own, sometimes more than one when one would suffice (e.g., line 53, page 2; line 114 in page 3, etc).

2. Also, there’s some inconsistency along the text. It is not clear why authors start by centering solely on mammals, omitting yeasts and plants data obtained previously than the one for mammals (e.g., oligomerization of PEX11), and after describing how peroxisome growth and division occurs, start including yeasts/plants data. 

Suggestions/corrections/questions:

1. lines 135-137, page 4: “PEX11β possesses two membrane-spanning domains with a very short C-terminus and a larger N-terminus, both facing the cytosol (Fig. 2). Please provide references (e.g., your ref 53, Koch and Brocard 2012 JCS)

2. line 138, page 4: Ref. 39 is wrong. Please add: Su J et al 2018 BBA Biomembr (DOI: 10.1016/j.bbamem.2018.02.029 ). 

3. line 139, page 4: “Furthermore, the N-terminus is required for oligomerisation of PEX11β [40,41].” The reference to the paper of Kobayashi S. is missing (Kobayashi S et al 2007 Exp Cell Res (DOI: 10.1016/j.yexcr.2007.02.028 )).

4. Fig.2, page 5: FFAT should be described in the figure legend.

5. In line 210, page 6: “Like FIS1 and MFF, MIRO1 is also a C-tail anchored membrane protein with dual peroxisomal and mitochondrial localization [34].” The ref. 31 (Covill-Cooke C 2020 EMBO Rep) should also be cited here.

6. lines 220-221, page 6: “PEX5 is a cytoplasmic import receptor for peroxisomal matrix proteins, which can interact with the peroxisomal membrane to deliver cargo proteins.” Please add (a) reference(s). 

7. lines 224- 227, page 7 : “It results in import-deficient peroxisomes, which lack peroxisomal enzymes within the matrix and are metabolically inactive. However, peroxisomal membranes can still be formed, but peroxisomes are reduced in number and enlarged, presenting as “ghosts” (empty peroxisomal membrane structures).” References are missing.

8. line 232, page 7: “acyl CoA oxidase (ACOX),” it should be instead “acyl-CoA oxidase I (ACOX1)”

9. Definitions of “FFAT-like motif”,“MSP domain”, “MAVS”, “GDAP1”, “VAPB”, “elovl1” , “PUFA”, “LPC”, “CNS”, “AAV”, “FA”, are missing, please define in the text the first time they are used.

10. Lines 331-334, page 9: “Therefore, it is likely that differences observed in peroxisome morphology between patients (and even between different studies on the same patient cells [98,100]), could be due to differences in the cell type/cellular environment rather than the specific mutation.”. Authors can’t say that specific mutations cannot cause differences in peroxisome morphology, because different mutations even when causing just a single amino acid substitution can have different functional consequences, and therefore may justify the different morphology. For example, a missense mutation in the GTPase domain of DRP1 can completely block GTP hydrolysis – e.g., by abolishing GTP binding – or could just reduce its binding affinity and the outcomes are completely different. Therefore, I suggest to change the sentence. Probably something like: “Therefore, it is likely that differences observed in peroxisome morphology between patients (and even between different studies on the same patient cells [98,100]), may also arise/could also be due to differences in the cell type/cellular environment.”

11. lines 339-342, page 9: “Unlike peroxisome biogenesis disorders (PBDs), patients with DRP1 mutations do not typically show an increase in serum VLCFA levels [96,99,101], or patient fibroblasts do not show defects in peroxisomal β-oxidation [89], indicating that the elongated peroxisomes can still perform their usual metabolic functions.” Please correct to: “Unlike patients with peroxisome biogenesis disorders (PBDs), those with DRP1 mutations (…) VLCFA levels [96,99,101], nor their fibroblasts show…”

12. Line 429-430, page 11: ”However, recent studies with MFF-deficient skin fibroblast have revealed that the hyper-elongated peroxisomes can be still degraded when autophagy/pexophagy is triggered.”. Please correct “fibroblasts”.

13. Lines 739-740, page 19: “The model is also applicable to other disease conditions, e.g. the loss of PEX5 in Zellweger spectrum disorders [29].” It is not clear to me what the authors meant. Please explain.

14. The definition of PPAR is misplaced. It is provided in line 819, page 19, yet the first time it is used is in line 748, page 17.

15. In Appendix A, page 22: There’s inconsistency in the definition of DRP1, in the main text it is defined as Dynamin 1-related protein (DRP1), here, the same acronym is used for Dynamin 1-like protein (when it should be DLP1). 

16. In Appendix A, page 22: First line of the table, in “Organelle Alterations”, “Defective fission of mitochondria and peroxisomes, slightly elevated VLCFAs but no”. The rest of the sentence is missing. 

Other minor issues:

1. “in vitro” and “de novo” are not always in italic, 

2. “see section” only sometimes in bold

3. Ref. 46 has extra text in the doi “http”

4. e.g. should be in italic and contain a coma afterwards “e.g.,”

5. lines 258-260, page 7: “Recently, a role for VPS13D 258 in peroxisome biogenesis has been reported [70]. VPS13D can interact with MIRO1 and 259 VAPB to form a bridge between the ER and peroxisomes [71]).” There’s an extra bracket (the round one), please remove.

6. Line 277, page 7: “AtPEX11a” It should be an alpha character at the end.

7. lines 339-342, page 9: it should be degeneration

8. Lines 429-431, page 11: “However, recent studies 429 with MFF-deficient skin fibroblast have revealed that the hyper-elongated peroxisomes 430 can be still degraded when autophagy/pexophagy is triggered [50].” It should be read “fibroblasts”.

9. line 621, page 15: “whereas ACBD5 overexpression decreased long dis-620 tance movements in hippocampal primary neurons (Wang et al. 2018)”. This reference is not listed in the references section.

10. Lines 744-746, page 17: “Peroxisomes are highly dynamic and plastic organelles, which can adopt their num-744 ber and morphology to changes in the environment (see section 1.) (reviewed in 745 [15,157,158].”. There’s a round bracket missing at the end of the sentence.

11. Page 36: Ref.119 is incomplete, issue, volume, pages and doi are missing.

12. Oligomerization and oligomerisation are both used (text vs table) .

Author Response

Comments and Suggestions for Authors:

In this paper, Carmichael RE and colleagues provide an extensive and very complete review on peroxisomal membrane dynamics particularly focusing on peroxisome biogenesis according to the growth and division model. The role of different division associated proteins in these processes, as well as the role of peroxisome-ER contacts in lipid transfer for peroxisomal membrane proliferation, is fairly discussed, and additionally, the authors provide a sum up of all the published data regarding clinical phenotypes and pathophysiology of patients with defects in these proteins and raise potential ways for the treatment of these disorders. I think the review is very well written and would be of great interest to readers of the journal and therefore I recommend its publication after some minor corrections.

We are grateful to the reviewer for their enthusiastic appraisal of our manuscript and have addressed their comments below.

General comments:

  1. Some references are missing, while some are redundant. Authors should be careful when choosing the papers to cite. When review papers are needed to support a sentence, they tend to cite only their own, sometimes more than one when one would suffice (e.g., line 53, page 2; line 114 in page 3, etc).

We have considered those suggestions and revised where appropriate, but would also like to indicate that the review articles cited cover different aspects of peroxisomal growth and division, which are not covered in only one of those articles.

  1. Also, there’s some inconsistency along the text. It is not clear why authors start by centering solely on mammals, omitting yeasts and plants data obtained previously than the one for mammals (e.g., oligomerization of PEX11), and after describing how peroxisome growth and division occurs, start including yeasts/plants data. 

We are focussing on mammalian evidence given this is a patient-focussed paper, but include data from yeast/plants where it is of interest, e.g. to provide molecular and functional insights.

We also thank the reviewer for their helpful suggestions/corrections/questions (listed below), which we have revised in the text (see Track Changes/highlighted references). The only exception is Minor Issue #6 –AtPEX11a is correct as it is, since the PEX11 isoforms in plants are designated with Latin characters (not Greek as in mammals).

Suggestions/corrections/questions:

  1. lines 135-137, page 4: “PEX11β possesses two membrane-spanning domains with a very short C-terminus and a larger N-terminus, both facing the cytosol (Fig. 2). Please provide references (e.g., your ref 53, Koch and Brocard 2012 JCS)
  2. line 138, page 4: Ref. 39 is wrong. Please add: Su J et al 2018 BBA Biomembr (DOI: 10.1016/j.bbamem.2018.02.029 ). 
  3. line 139, page 4: “Furthermore, the N-terminus is required for oligomerisation of PEX11β [40,41].” The reference to the paper of Kobayashi S. is missing (Kobayashi S et al 2007 Exp Cell Res (DOI: 10.1016/j.yexcr.2007.02.028 )).
  4. Fig.2, page 5: FFAT should be described in the figure legend.
  5. In line 210, page 6: “Like FIS1 and MFF, MIRO1 is also a C-tail anchored membrane protein with dual peroxisomal and mitochondrial localization [34].” The ref. 31 (Covill-Cooke C 2020 EMBO Rep) should also be cited here.
  6. lines 220-221, page 6: “PEX5 is a cytoplasmic import receptor for peroxisomal matrix proteins, which can interact with the peroxisomal membrane to deliver cargo proteins.” Please add (a) reference(s). 
  7. lines 224- 227, page 7 : “It results in import-deficient peroxisomes, which lack peroxisomal enzymes within the matrix and are metabolically inactive. However, peroxisomal membranes can still be formed, but peroxisomes are reduced in number and enlarged, presenting as “ghosts” (empty peroxisomal membrane structures).” References are missing.
  8. line 232, page 7: “acyl CoA oxidase (ACOX),” it should be instead “acyl-CoA oxidase I (ACOX1)”
  9. Definitions of “FFAT-like motif”,“MSP domain”, “MAVS”, “GDAP1”, “VAPB”, “elovl1” , “PUFA”, “LPC”, “CNS”, “AAV”, “FA”, are missing, please define in the text the first time they are used.
  10. Lines 331-334, page 9: “Therefore, it is likely that differences observed in peroxisome morphology between patients (and even between different studies on the same patient cells [98,100]), could be due to differences in the cell type/cellular environment rather than the specific mutation.”. Authors can’t say that specific mutations cannot cause differences in peroxisome morphology, because different mutations even when causing just a single amino acid substitution can have different functional consequences, and therefore may justify the different morphology. For example, a missense mutation in the GTPase domain of DRP1 can completely block GTP hydrolysis – e.g., by abolishing GTP binding – or could just reduce its binding affinity and the outcomes are completely different. Therefore, I suggest to change the sentence. Probably something like: “Therefore, it is likely that differences observed in peroxisome morphology between patients (and even between different studies on the same patient cells [98,100]), may also arise/could also be due to differences in the cell type/cellular environment.”
  11. lines 339-342, page 9: “Unlike peroxisome biogenesis disorders (PBDs), patients with DRP1 mutations do not typically show an increase in serum VLCFA levels [96,99,101], or patient fibroblasts do not show defects in peroxisomal β-oxidation [89], indicating that the elongated peroxisomes can still perform their usual metabolic functions.” Please correct to: “Unlike patients with peroxisome biogenesis disorders (PBDs), those with DRP1 mutations (…) VLCFA levels [96,99,101], nor their fibroblasts show…”
  12. Line 429-430, page 11: ”However, recent studies with MFF-deficient skin fibroblast have revealed that the hyper-elongated peroxisomes can be still degraded when autophagy/pexophagy is triggered.”. Please correct “fibroblasts”.
  13. Lines 739-740, page 19: “The model is also applicable to other disease conditions, e.g. the loss of PEX5 in Zellweger spectrum disorders [29].” It is not clear to me what the authors meant. Please explain.
  14. The definition of PPAR is misplaced. It is provided in line 819, page 19, yet the first time it is used is in line 748, page 17.
  15. In Appendix A, page 22: There’s inconsistency in the definition of DRP1, in the main text it is defined as Dynamin 1-related protein (DRP1), here, the same acronym is used for Dynamin 1-like protein (when it should be DLP1). 
  16. In Appendix A, page 22: First line of the table, in “Organelle Alterations”, “Defective fission of mitochondria and peroxisomes, slightly elevated VLCFAs but no”. The rest of the sentence is missing. 

Other minor issues:

  1. “in vitro” and “de novo” are not always in italic, 
  2. “see section” only sometimes in bold
  3. Ref. 46 has extra text in the doi “http”
  4. e.g. should be in italic and contain a coma afterwards “e.g.,”
  5. lines 258-260, page 7: “Recently, a role for VPS13D 258 in peroxisome biogenesis has been reported [70]. VPS13D can interact with MIRO1 and 259 VAPB to form a bridge between the ER and peroxisomes [71]).” There’s an extra bracket (the round one), please remove.
  6. Line 277, page 7: “AtPEX11a” It should be an alpha character at the end.
  7. lines 339-342, page 9: it should be degeneration
  8. Lines 429-431, page 11: “However, recent studies 429 with MFF-deficient skin fibroblast have revealed that the hyper-elongated peroxisomes 430 can be still degraded when autophagy/pexophagy is triggered [50].” It should be read “fibroblasts”.
  9. line 621, page 15: “whereas ACBD5 overexpression decreased long dis-620 tance movements in hippocampal primary neurons (Wang et al. 2018)”. This reference is not listed in the references section.
  10. Lines 744-746, page 17: “Peroxisomes are highly dynamic and plastic organelles, which can adopt their num-744 ber and morphology to changes in the environment (see section 1.) (reviewed in 745 [15,157,158].”. There’s a round bracket missing at the end of the sentence.
  11. Page 36: Ref.119 is incomplete, issue, volume, pages and doi are missing.
  12. Oligomerization and oligomerisation are both used (text vs table) .